# PATROL: Participatory Activity Tracking and Risk Assessment for Anonymous Elderly Monitoring

**DOI:** 10.3390/s22186965

**Published:** 2022-09-14

**Authors:** Research Dawadi, Teruhiro Mizumoto, Yuki Matsuda, Keiichi Yasumoto

**Affiliations:** 1Graduate School of Science and Technology, Nara Institute of Science and Technology, Ikoma 630-0192, Nara, Japan; 2Graduate School of Information Science and Technology, Osaka University, Suita 565-0871, Osaka, Japan

**Keywords:** elderly monitoring, successful aging, mobile application, gerontechnology

## Abstract

There has been a subsequent increase in the number of elderly people living alone, with contribution from advancement in medicine and technology. However, hospitals and nursing homes are crowded, expensive, and uncomfortable, while personal caretakers are expensive and few in number. Home monitoring technologies are therefore on the rise. In this study, we propose an anonymous elderly monitoring system to track potential risks in everyday activities such as sleep, medication, shower, and food intake using a smartphone application. We design and implement an activity visualization and notification strategy method to identify risks easily and quickly. For evaluation, we added risky situations in an activity dataset from a real-life experiment with the elderly and conducted a user study using the proposed method and two other methods varying in visualization and notification techniques. With our proposed method, 75.2% of the risks were successfully identified, while 68.5% and 65.8% were identified with other methods. The average time taken to respond to notification was 176.46 min with the proposed method, compared to 201.42 and 176.9 min with other methods. Moreover, the interface analyzing and reporting time was also lower (28 s) in the proposed method compared to 38 and 54 s in other methods.

## 1. Introduction

Advancements in medicine and health care technologies have led to an increase in life expectancy over the years. It is expected that, by 2050, there will be at least 2 billion people over the age of 60 years [1]. The statistical handbook of Japan released in 2021 by the Statistics Bureau, Ministry of Internal Affairs and Communications, Japan has revealed that, in 2015, there were about 22 million households with residents aged 65 and above, including 6 million who lived alone [2]. Living independently, especially for the elderly, is risky because, in addition to mental problems such as memory loss, depression, and loneliness, there can be physical problems such as falling down, issues with eyesight, hearing loss, back pain, etc. [3]. Though different remedies have been developed for different types of physical and mental ailments, with an increasing number of elderly people, it is apparent that there is a need for monitoring and anomaly detection mechanisms. A lot of research has thus contributed to recognizing, predicting, and monitoring activities inside smart homes [4,5].

As people get older, their involvement in different physical and mental activities decline [6]. They go out less, engage in activities related to physical fitness less, have difficulty with reading for a long time due to weakened eyesight, and so on. Similarly, they deal with issues they had not dealt when they were younger, such as the need to take medication every day and the adverse effects of missing a meal. Similarly, falls or any similar incidents tend to make the elderly cautious in their activities, impacting their confidence, activity completion, and social interactions. Therefore, it becomes imperative to track whether the elderly has completed basic day-to-day activities every day in order to detect any abnormal conditions that might have occurred or might occur [5,7]. There have been many advancements in human monitoring, collecting vital health statistics and tracking human behavior over the recent years [8]. Off-the-shelf sensors can be now used in houses that can provide information about light intensity, temperature, and usage of doors and appliances of houses [9], making it possible to determine activities inside the house.

Research has also been carried out in health care centers, but implementing such technology in the home environment is more suitable for the elderly. The elderlies have made memories over the years in their home and have possessions they cherish [10]. Hence, they feel more comfortable to live in their own home as well as conduct their basic everyday activities. Moreover, hospitals and health care centers are either expensive or overbooked. The cost can be reduced by up to 52% when patients receive treatment and help in their home compared to hospitals [11]. It is therefore necessary to develop systems that can help to enhance elderly care in their own home rather than hospitals or support homes. Professional caretakers are expensive as well, and with the increasing number of elderly people they tend to be overbooked and busy [5]. Home monitoring technologies can help family members and relatives who are far away be assured about the safety and contentment of the elderly [1]. However, their busy schedule may not allow them to monitor the activities regularly, which is why personnel dedicated to remote monitoring such as remote caretakers or volunteers should be assigned the monitoring responsibilities.

With these issues in consideration, in this paper, we propose a monitoring system, PATROL (Participatory Activity Tracking and Risk assessment for anOnymous eLderly monitoring) that can track basic activities of the elderly anonymously inside their home and detect or prevent any potential risks in their day to day activities using a smartphone application. For the successful implementation of the PATROL system, the following requirements need to be fulfilled: *(Req. 1) anonymous monitoring*, *(Req. 2) timely monitoring and report of activities*, and *(Req. 3) easy and intuitive risk detection* because of the following reasons.

Home monitoring can be considered intrusive as in some cases, the elderly may prefer to hide things in their house if there is a video based monitoring or surveillance system [12]. Similarly, they are also usually concerned about privacy and security, and the types of information about them that are disclosed [1]. This is why we propose anonymous monitoring (Req. 1), where any personal details of the elderly being monitored is not disclosed to the monitoring person. Smartphones are a suitable device for regular tracking and monitoring since many people carry them the whole day or they are always in the vicinity of the users. Furthermore, notifications have become an essential feature of most of the smartphone applications [13]. This is why we propose a smartphone application that can be used by volunteers for tracking and monitoring activities of elderly people. Similarly, we send frequent notifications in the smartphone application, which ensures that the monitors can quickly access information about the activities of the elderly, compared to using web pages (Req. 2). Continuous usage of smartphone applications in general has been attributed to factors such as ease of navigation, ease of carrying out actions within the application, and appropriate visual clues [14], which is why we focus on the visualization of activities and propose a method for visualizing activities and detecting risks in the daily activities that not only helps to identify risks in the activity visualization easily, but also incurs a lesser burden to the monitoring person (Req. 3).

Therefore, in this paper, we propose an elderly monitoring system that can be used by anonymous volunteers to check everyday activities of the elderly and determine if there are any risky situations in their day to day activities. The anonymity is maintained by not disclosing any personal or private information of the elderly to the volunteers, and similarly by not disclosing any personal or private information of the volunteers to the elderly person. Using volunteers for elderly care is a very common practice in Japan [15] where part-time civil servants committed by the Minister of Health, Labor, and Welfare as volunteers, locally known as minsei-iin, are assigned to regularly check the elderly people personally, have a conversation with them, etc. These part-time civil servants are people who volunteer themselves in the area of helping children, elderly people, people with disabilities, etc. and have no mandatory obligation to serve in such areas. We believe that our system is an extension of such practice in the field of elderly care. Instead of visiting the elderly, our volunteers can check the elderly by using the smartphone application even if they are not in the vicinity of the elderly. This is helpful in cases when the elderly might not prefer an unknown person to visit them personally, and also in cases where the number of people serving as minsei-iin might not be enough. Since in our system, we aim to use multiple monitors, we ensure that the activities of the elderly are regularly checked. To maintain anonymity, even if the volunteers discover a risky situation in the daily activities of the elderly, the handling of such a situation, in person, is carried out by emergency contacts of the elderly, and not the volunteers themselves. For our system, we define risk as a deviation in start/end time and duration of activities from the usual routine of the elderly people.

We developed an Android based smartphone application that provides information about the completion of seven basic activities: sleep, shower, medication, breakfast, lunch, dinner, and entertainment (use of television (TV)). We created a dataset by including some risky situations in the elderly activity dataset [16] to determine if those situations can be detected using our application design. To make the monitoring process less burdensome and intuitive, we also included visualization features such as a candlestick chart representation of activities, single interface design, and textual and color codes for their current state, through which it is easy to infer any deviation in the completion time and duration of activities. Similarly, we focused on quick tracking and monitoring of activities by including two types of notifications to trigger frequent use of the smartphone application: one sent every two hours, and another sent immediately after the elderly completed an activity.

The main contributions of this paper are the following:First, we proposed a novel system that can be used by volunteers to anonymously monitor completion of daily activities of elderly people, and report if they detect any deviation in the activities compared to the usual routine of the elderly. We developed an Android based smartphone application that is designed with numerous visualization features and two types of notification strategies to make activity monitoring and detection of anomalies easy, intuitive, quick, and less burdensome.Second, we evaluated our smartphone application with visualization features and a two notification strategy by comparing it with baseline methods (the method without the notification strategy or the visualization features) and confirmed that our proposed method not only provided better risk identification, but also incurred lesser burden on the monitoring person. We also show that our proposed method resulted in quick tracking and monitoring of activities.

The rest of the paper is organized as follows: Section 2 introduces some available research and how they relate to our study. In the next section, Section 3, we introduce our system followed by the explanation of our smartphone application. We explain the evaluation study and findings of the study in Section 4 and in Section 5, and we discuss the significance of the results for our system along with the limitations of this study. Finally, we conclude with our contributions in Section 6.

## 2. Related Studies and Challenges

Increasing demands in safe, secure, and smart homes for the elderly have led to many research and advances in the field of home monitoring and home automation [4,5,17]. Similarly, with increasing use of smartphone notifications to provide various information to users, we look into studies that explored reliable triggers to inspire people to respond early to mobile notifications. With these factors in mind, we studied existing research, which are divided into two subsections that deal with activity detection and remote monitoring, and importance of smartphone notifications.

### 2.1. Activity Recognition and Remote Monitoring

In recent years, research dedicated to monitoring people and their activities inside their house has been increasing rapidly since activities of people can be identified with the help of various sensors that can be attached to different household objects [18]. Most home monitoring methods utilize camera or video captures to learn about the activities of the elderly [7]. Video and microphone based monitoring can be time consuming for monitoring, burdensome, and also intrusive [12], and also restrict the area of the house the elderly can occupy to regular monitoring [5]. Numerous research studies have been carried out to tackle not only such problems, but also improve recognition accuracy and reduce the burden of using wearable sensors. The daily activity pattern of elderly people was identified using only motion and domotic sensors by identifying the duration of occupancy of a certain room by the elderly [1]. Similarly, using energy harvesting PIR (passive infrared sensor) and door sensors, an activity recognition system was developed that was efficient as well as cost effective [19].

Many other activity recognition systems utilise non-wearable sensors such as motion sensors [20], Bluetooth Low Energy (BLE) beacon [4,17], wireless accelerometers [21], a combination of temperature, humidity, and illumination sensors [22], and a combination of ECHONET Lite appliances and motion sensors [8]. Similarly, deploying a system that used motion sensors, environmental sensors, and a button to be pressed at the start and end of an activity, daily activities of the elderly were collected for a period of about two months in houses consisting of elderly people [16]. All these studies help to highlight that it is possible to collect activities in the house using sensors such as motion sensors, environmental sensors, etc. accurately without the use of any wearable sensors in a cost-effective way and handling concerns for privacy and security.

Activity recognition systems also allow the elderly to live an independent life in their own house whilst their activities are monitored remotely [5]. There have been measures to monitor vital signs and biomedical signals of adults with medical conditions [23] or people working in extreme conditions such as firefighters [24]. The Allocation and Group Awareness Pervasive Environment (AGAPE) system used on-body sensors to monitor the elderly and contacted nearby caregiver groups in case it detected an anomaly in sensor data [25]. Systems can also contact the emergency contact, or caregivers for the elderly if any anomaly in the collected data are observed, for example, when the data exceed a predefined threshold [26,27]. When it comes to elderly remote monitoring, fall recognition systems are also very important, with some systems recording the average response time of fall detection between 7 min and 21 min [28]. The systems can detect falls using various types of sensing strategies such as acoustic sensors [29], wearable sensors [30], or accelerometers in smartphones [31].

Many commercially available products are also available that are used to monitor the elderly remotely. Systems such as Mimamori [32] and Canary [33] are specially designed to monitor activities of elderlies by their children and close family members who live in a distant location. Another system, GreatCall Responder, uses a physical button, called a responder, that the elderly can press in case they feel they have an emergency, and the system contacts their caregiver [34]. Similarly, there are systems that track numerous activities using motion sensors that remote caregivers can monitor using a private and secure webpage [35,36]. There are also systems that include secure video communication between doctor and patients for regular or emergency situations, remote health monitoring, and emergency care services [37,38].

Many elderly people, however, regard new technologies as an invasion of their privacy and security [10], and tend to accept technologies only if it is beneficial to them or it adheres to their day to day activities without providing any hindrance [39]. A study revealed that being monitored in their house, conducting their day to day activities did not affect regular daily behavior of the elderly [40]. Their extensive study requested the elderly to answer online questionnaires weekly and included daily activities of sending, reading and deleting emails, along with tracking their total everyday activities, walking speed, and time spent outside their home. Hence, if issues of privacy and security are tackled, and the elderly feel that the activity recognition system will be valuable to them, then there is higher chance of acceptance of such a system.

These systems also provide some areas of concern. The alerts are sent to caretakers of health professionals via text or email [28] or direct phone calls [26]. However, the number of false alarms, which can be as high as 5 in one hour [29], can cause annoyance to the caretakers. Similarly, even though the accuracy of fall detection systems is high such as 97.5% [28] or 94% [30], the information regarding the time it takes such systems to inform the caretaker or the time it takes caretakers to respond are not explicitly evaluated. In another system, the activities of elderly were divided into critical, stable, scheduled and overlooked, and alerts for them were generated in a smartphone application as per the type such as after 5 min of usual time for critical activity such as medication and after 30 min for other activities [41]. These alerts were first sent to the elderly, and if they failed to respond, the caretakers were alerted. However, it is difficult to determine the exact time the elderly might prefer to do their daily activities. Similarly, in the case of emergency, the elderly may not be physically able to respond to alerts [41] or press the emergency button [34].

### 2.2. Smartphone Notifications

Smartphones have become a daily necessity as it helps to tackle isolation, as well as helping to stay in contact with family and friends easily [42]. Smartphones have become an essential tool to be updated about personal health, work, and news updates [43]. Smartphone owners interact with their phones an average of 85 times a day [44] which makes them a befitting tool for remote monitoring. Notifications are essential to keep the users updated about news, emails from work, and information from social media [45]. Although initially they were intended for short message services (SMS) or emails, these days, notification features are used by almost all of the applications to attract attention of the users. A study determined that notifications can be divided into two categories: personal notifications like emails, SMS, or those from social networking sites; and mass notifications like news and advertisements [46]. They concluded that people tend to attend to personal notification faster and more frequently than mass notifications.

The response to notifications depends on different factors such as sender, type of alert, and the visual representation of the alert [47]. In a recent study, it was shown that users receive approximately 64 notifications each day [48], hence the context of a notification plays an important role in the response of the notification. Time of notification reception, activeness of the user, and amount of time the user will take to respond to the received notification are influential for opening the notification promptly [46]. From a study of about 200 million notifications from more than 40,000 users [13], it was discovered that users view each notification differently and prefer to respond to notifications from social networking sites quickly over those from the smartphone system or emails.

Notifications can however lower task performance and affect attention of the user negatively [45]. Response time and response rate of notifications were determined by analyzing the current context of the user through audio from their smartphones [49]. They concluded that the present context of the user plays a very vital role in the response time as well as response rate of the notifications. Similarly, a systematic review on the effects of context aware notification management systems found that context aware notifications increase the response rate [50]. However, it is difficult to predict what time and context can be considered as appropriate for interruption. Since remote monitoring technologies can send multiple notifications in a day, it is essential to determine if such notifications will be viewed as disruptive. Similarly, to our knowledge, the effectiveness of smartphone notifications in remote monitoring systems, especially using multiple types of notification strategies, has not been investigated.

### 2.3. Challenges

We found out that there are many methods with which activities can be detected accurately. However, in the case of elderly people, it is also necessary to monitor such activities on a regular basis [5]. A smartphone application, equipped with adequate notification strategies, can provide a quicker remote monitoring compared to most of the remote monitoring platforms that are currently web based [35,37,38]. The smartphone application that we have designed can be used to instantly monitor completed activities and receive quick feedback from the monitoring person. It is essential not only to track activities, but also check if any risks that have occurred, and predict or prevent any potential risks in the daily life of elderly. Hence, at first, it is necessary to determine what activities to monitor and if those activities can be properly visualised in the application, and, furthermore, if any deviation in the routine of the elderly can be distinguished so that any potential risky situation of the elderly can be detected. Similarly, it is essential to identify if using the application, and monitoring activities regularly will put a burden on the monitoring person. With all this in mind, we propose the following research questions (RQs), which we try to verify with an experimental study:RQ1: Is it possible to identify daily routine of individuals using a smartphone application?RQ2: Can a monitoring person detect potential risks in day to day activities based on visualization of activities in our application?RQ3: Is constant notification and using the application a burden for the monitoring person?

## 3. System Design

In this section, we first explain the overview of the proposed PATROL (Participatory Activity Tracking and Risk assessment for anOnymous eLderly monitoring) system. Then, we describe the design and interface of our smartphone application in detail.

### 3.1. System Overview

The architecture of PATROL system is shown in Figure 1, where we denominate the elderly being monitored as *Target* and the person conducting monitoring as *Monitor*.

The monitoring can be conducted in different ways. One Target can be monitored by a single or multiple Monitors and one Monitor can conduct monitoring of a single or multiple Targets. Consequently, multiple Monitors can be used to monitor multiple Targets.

The overall system can be further divided into four sections: activity recognition, monitor generation, notification generation, and smartphone application, as highlighted in Figure 1. In this research, we focus mainly on the two sections: notification generation and smartphone application. We will now discuss each of the sections and their application in our overall system.

#### 3.1.1. Activity Recognition

Most elderly people have a definite time and duration for their activities, and follow a routine set of activities throughout the day [51]. It is important to check for everyday basic activities because, with old age, these important basic daily activities can sometimes be missed or incomplete or not properly carried out [7]. For the purpose of our research, we assume that the Target is residing in a smart home equipped with an activity recognition system, where it is possible to collect information related to daily activities like eating, sleeping, watching TV, taking medicine, etc. through the use of different kinds of sensors and power consumption meters available in the house [1,8,16]. We have designed our system in a way that it can incorporate any available activity recognition systems. Therefore, it is easy to integrate in houses which already have an activity recognition system. Activities that we showcase in the smartphone application are shown in Table 1. We believe that the state of everyday basic activities can be used as criteria to determine the wellness of the elderly person. There can be instances when anomalies can occur whilst conducting activities that are not listed in Table 1. However, such incidences will subsequently impact the occurrence of basic activities that we aim to monitor. Therefore, our system can detect anomalies that can occur doing activities that are not directly monitored in our application. Since our aim is to disclose as less information about the Target as possible, whilst making it possible to determine their current status, we only use time of completion and duration of the activities to provide information about them. We assume that the activity recognition system outputs events (i.e., start and end times of activities performed by the resident) which are utilized for data visualization and notification generation, as shown in Figure 1. This feature will be further discussed in Section 3.2.

#### 3.1.2. Monitor Generation

The PATROL system is designed to be used especially for monitoring the elderly, and to be deployed in nursing homes, elderly residential areas, care homes, municipalities, etc. The overall system needs to be handled by a system administrator who can be the head of the residence association or personnel who work in such institutions. In case of changes in the system administrator, then the outgoing system administrator under the authority of the local welfare committee (and/or residents’ association) will have to train the new system administrator immediately. In our context, Monitors are usually volunteers who work in the field of helping elderly in care homes, elderly residential areas, etc. The Monitors participate in tracking the activities and determining risky situations in the activities of the elderly. The system administrators have the responsibility of training the Monitors to use the smartphone application, assigning Monitors for each Target, assessing the performance of Monitors and determining if any change needs to be done. In case of changes in Monitors as well, the training of new Monitors is handled by the system administrators. Similarly, the initial testing and assessment of our application is handled by the system administrators as well who check if the system is working properly, and the application is generating activity reports and notifications regularly. Since our application shows activities not just of the current day, but of a period of days (e.g., week), including previous days, a new user can still be familiar with start/end and duration of activities of a range of days and deduce a pattern or routine of the target easily.

The number of Targets assigned for each Monitor may vary based on the preference of each volunteer. The volunteers are free to choose a minimum or maximum number of Targets to monitor, after which the system administrator will assign them Targets. Therefore, the number may vary from a single Target to multiple ones based on each volunteer.

#### 3.1.3. Notification Generation

To encourage regular usage of the application, frequent notifications are sent to the Monitors. This functionality helps to timely track the recent activities of the Target and detect any change in the usual routine. We think there should be two types of notifications generated: emergency and general. General notifications are sent to remind monitors about using the application and check current activities of the target. Emergency notifications are sent when the system itself detects abnormalities in the recent activities of the target. We do not generate or analyze emergency notifications in this research because we aim to determine how often general notifications are responded by the Monitors, if they motivate the monitors to frequently use the application or not, and if constant notifications will be burdensome or disturbing.

The notification scheduling techniques that are commonly used can be divided into three types: randomized time points in a day, timed at specific intervals, and event dependent times [52]. In our system, general notifications are generated by using two types of notification strategies: timed at specific intervals and event dependent notifications. This ensures that the monitors are notified regularly to use the application, and can instantly check information about the activity completed.

#### 3.1.4. Smartphone Application

The information collected from the house of the Target is utilised to create graphical representation of activity completed in a time series form which helps to identify a pattern in the time of completion of activity and its duration, so that any deviation from the usual pattern can be identified with ease. Hence, we develop an Android based smartphone application, PATROL, which can be used to view the activities completed and send reports. Since smartphones have become a common gadget among the elderly as well [53], our application can be used by the young volunteers as well as the elderly. For our research, we conduct an experiment using smartphones, but the application can also be used in any other Android based devices like tablets.

The interaction between the Monitor and the application is shown in Figure 2. We have tried to minimize the number of actions required to be carried out by Monitors. In the application, the Monitors receive notifications as a trigger so that they can check the time of completion and duration of the activity for the current day and previous days, after which they can judge whether the Target is in a risky situation or not, and submit a report. If the Monitor reports that the Target is in a high risk situation, then the application can notify the system administrator and emergency contacts of the elderly via text, email, or automated phone calls, who can take necessary actions immediately. The Monitors did not disclose any details of the Target even in such situations to maintain the anonymity of our system. The system administrators, who are in the vicinity of the Target, will take the responsibility for checking the Target as soon as such reports are received. The report sent by the Monitors are saved and analyzed to evaluate their monitoring capability.

For the accurate analysis of our application, it is necessary that risky situations of the Targets are identified correctly. We, however, at first need to define what these risks are, and how they can be related to real life situations. We created a total of four risk stages, as shown in Table 2. These risks are based on the changes in the routine of the Target. If there is no change in their routine i.e., no noticeable deviation in their activity, then we regard the risk as None. Low and Medium risks are defined based on the amount of deviation from the usual start/end time or duration of the activities. High risks refer to situations when the activity has not started, or completed indicating that the Target needs urgent attention.

We have used standard deviation to define *low* and *medium* level risks. We calculated standard deviation of duration and time of completion of each activity, for each targets. Then, we defined *low* and *medium* level of risks as follows:Low risk-duration ±1.5 × standard deviation of duration-time ±1.5 × standard deviation of activity completion timeMedium risk-duration ± 3 × standard deviation of duration-time ± 3 × standard deviation of activity completion time

The purpose of using this technique is that it gives us a wide range of duration and activity start/end times that we can relate with risks in real life scenarios. The low risk indicates that the deviation in time or duration was not so concerning, which meant that the elderly had some problems but were able to deal with them themselves. Medium risk indicates a higher deviation in time or duration of activity, which indicates that the elderly might not be doing so well and need to be attended to personally. Since we have ourselves defined these ranges of duration and start/end times for low and medium risks, they are flexible, and hence can be modified based on the activity data of the elderly.

### 3.2. Application Design

Even though recent technologies have been designed and developed targeting with an average young user in mind, who is efficient at handling new systems or devices [54,55], we have tried to make the interface simple and intuitive so that it can be used by people of all ages conveniently. As shown in Figure 2, the number of tasks to be carried out by the Monitor in the application are very minimal. Therefore, we believe that the application will be easy to use, and the burden of using the application will be low for the Monitors. Our final goal is to achieve remote elderly care and prompt identification of risky situations; however, we believe that to achieve them, the design and interface of the application should be favorable to the monitors. We aim for our concern of providing a continuous and detailed elderly care system, and an easy and intuitive interface for monitors does not remain mutually exclusive. The actions in the application to be carried out are: respond to notifications, check activity, and submit a report. Below, we will explain different features in the interface of the smartphone application, and the notification strategy that we developed.

#### 3.2.1. Features of the Application Interface

We have designed the application with various features in the interface that is aimed at helping the monitoring process. All the activities are shown in a single interface to reduce the burden of going back and forth between interfaces to monitor the activities. We will now discuss the features of the application interface.

##### Activity Report

The application shows the option to choose whom to monitor among a list of Targets, as shown in Figure 3a. Since our application is anonymous, the real names of the Targets are not shown. We used three commonly used names in Japan (Taro, Watanabe, and Yamazaki) to denominate the Targets in our application. Once the Target is chosen, then the activity report interface is shown, as shown in Figure 3b.

The activity report interface breaks down each activity into different cards, with each card showcasing the current status of the activity (incomplete, ongoing, or completed), activity completion time (in graph as well as text), and duration of the activity, as shown in Figure 3b,c. In case of activities like TV and medication that can occur multiple times in a day, each separate activity is represented by separate cards. The Candlestick chart style helps to identify a pattern in the time of completion of activity and its duration, so that any deviation from the usual routine can be recognized with ease. We use a candlestick chart to show activities because it can showcase the time as well as duration with clarity, and the difference between consecutive days is also understandable.

The Monitor, ideally, should be able to submit only one report per activity per day as well as provide the report for an activity only after the activity has been completed. Hence, in order to prevent multiple and erroneous reporting, we use two techniques: color codes in activity cards; and radio button for reporting. In the cases of activities that occur multiple times in a day (such as TV, medication), multiple activity cards of the same activity are shown. To avoid confusion for the users, only one activity card is shown at the start of the day, when no multiple activities have occurred. The activity cards are then subsequently added soon after their occurrence.

##### Colors Codes in Activity Cards

Traffic light colors have been used in various research studies, from labelling traffic colors on food to indicate their edibility or freshness [56,57], to using traffic colors as a means of self-monitoring by recording the weight and shortness of breath in a diary [58]. We use traffic color codes for the activity cards in order to make the current status of activities of the Target clear, as shown in Figure 4.

The background color of the activity card is represented by *red* when the Target has not completed the activity, as shown in Figure 4a. The current status information, shown as *Incomplete*, also gives an update that the activity has not been finished for the current day. The information about end time and the duration of the activity is also empty at this stage.

The background color of the activity card is represented by *red* when the target starts the activity, as shown in Figure 4b. The current status information is changed to *Ongoing* in this case, and the information about the start time of that activity is updated. The information about the duration of the activity is also empty at this stage.

The background color changes to *yellow* when the activity is finished by the Target. The current status is also updated, to *Complete*, along with information about end time and duration of the activity. Along with the change in color, the radio buttons for reporting the status are also shown below the card, as shown in Figure 4c.

When the Monitor reports about the activity, then the background color of the card is changed to *green*. Along with that, the radio buttons for reporting are hidden, as shown in Figure 4d. Thus, when the Monitor opens the application again after submitting a report, the option to report again is not available, and the color codes help them identify the activities they have already reported.

We believe that, since people are familiar with traffic colors and their functions, this feature in the application is intuitive, and helpful in clearly distinguishing the states of activity. The colors are also directly related to the state of the elderly as well as the necessity of Monitor’s attention. When the background color is red, activities are either ongoing or not started at all, which means that the elderly has not completed any activity. This state requires a higher amount of attention from the monitor because if the background color does not change from red for a prolonged time, then it should be deduced by the Monitors that the elderly might be in a risky situation and thus the Monitor should report, via an overall report card. When the background color of the card changes to yellow, it indicates that the elderly has completed an activity, and the monitor should now check the activity and submit a report. This state requires lower attention from the the Monitor compared to the red background color state. Similarly, a green color gives Monitors a confirmation that they have completed the reporting task already and should not pay any attention to that particular activity anymore.

##### Overall Report

Along with the activity cards for each activity, there is a separate card called Overall card. This, in general, is to report about overall impression about the status of the elderly. This can be reported multiple times by the Monitor throughout the day, and has the same reporting option of risks and confidence as in other activity cards, as shown in Figure 4d. Thus, when submitting reports for activities, the Monitor has the option to choose what they feel is the overall status of the Target based on their judgement of activities completed or not completed. In cases of High risk situations such as no activity or long deviation, the target will not register completion of activities regularly, which means that no notifications are sent and the activity cards are not updated. If no activity has been updated for a significant time, then the monitors can deduce that there is something wrong with the target. In such situations, they can report the emergency situation using the Overall card. The card also shows the type and time of previous response for the Overall card, to make it easier for the Monitor to recall their previous impression, as shown in Figure 5b.

##### Submit Report

The task for the Monitor is to check the activity report of the Target and analyze the information shown and then submit their report. The report can be submitted for one activity at a time, as well as for multiple activities at the same time. To submit the report for each activity, the Monitor needs to scroll down in the activity report interface and click the submit button at the end of the activity report interface as shown in Figure 3d.

If the monitor responds with *high* risk and *high* confidence to any activity, then the application can infer that the elderly might be in an emergency situation, and can promptly notify the emergency contact of the Target (friends, family or health professionals) via text message, email, or automated phone calls, and they can take necessary actions. Similarly, if more than two subsequent *medium* risks are reported with *high* confidence, then their emergency contact can be notified immediately. Thus, to provide a base to analyze the confidence of the report, we divided the confidence level for each report as *Low*, *Medium* and *High*, as seen in Figure 4c. The confidence levels hence act as reference points of risks for each activity, especially when there are multiple Monitors. The confidence level provides a perception of each of the Monitors and their report, and also helps to analyze their monitoring capabilities.

#### 3.2.2. Notification Strategy

We deploy two kinds of notification patterns in our application: recurring notifications (rN) and activity based notification (abN). We send notifications every two hours (rN) to provide a trigger to the targets to use the application. The period for recurring notification is two hours because we feel that two hours is an appropriate time gap for reminding users, as sending a notification every 30 min or an hour will be too disruptive. Analyzing the activity completion times and usual gap between activities, we feel that two hours is an appropriate gap to send a recurring notification. We have also analyzed the perception of users towards recurring notifications of two-hour intervals, and have empirically proved that they are not perceived as disturbing and were responded to about 87% of the time [59].

Apart from this, we also send a notification, abN, which is sent as soon as a target completes an activity. We mentioned in Section 2.2 that it is necessary to provide contextual information in notifications for quick responses. We provide the name of the target and the activity completed in the notification, to provide context of the notification to the monitors, as shown in Figure 6. To make distinction between the two types of notifications, we indicate abN with a red icon of notification (see Figure 6a) and rN with a blue icon (see Figure 6b).

## 4. Implementation and Evaluation

In this section, we will explain the details of the experiment conducted to analyze the application, including the dataset used for the application, multiple versions of PATROL application that we created, and finally explain the result of our study.

### 4.1. Multiple Versions of PATROL Application

In order to concretely determine that our proposed method of a graphical interface (GI), as shown in Figure 3b, is intuitive and has a higher degree of user acceptance, we needed to compare that interface with commonly used activity representation techniques. To make that distinction, we created a separate version of our application where activities were shown in a textual interface, rather than graphs. Figure 7 shows the activity report interface of this kind of version of the application. All the features of the application mentioned in Section 3.2 are included in this version as well, so the working principle is the same regardless of the interface. This helps create less confusion for the participants and ensures that the performance and perception of users is solely based on the type of interface, and not on other features of the application.

Similarly, we created a third version of our application (GR), in which we did not send notifications to the monitors when the activity was completed by a target. We only send them recurring notifications every two hours. With this version of the application, we aim to determine if the monitors are able to report about activities of the elderly even if they do not receive activity based notifications (abN) and thus our strategy of providing both abN (activity based notification) and rN (recurring notification) can be effective to encourage and motivate monitors to use the application frequently and receive continuous reports of activities of the target.

Table 3 summarizes the three versions of the application created, and we will use the same label for versions (GAR, TAR, and GR) in future discussions. GAR refers to the proposed version of PATROL, which consists of a Graphical interface, Activity based notification, and Recurring notification. We investigate the accuracy of risk identification, and the burden of use of our application by comparing the versions GAR and TAR (Tabular interface, Activity based notification, and Recurring notification). Similarly, we compare the effectiveness of using activity based notifications by comparing GAR with GR (Graphical interface and Recurring notification).

### 4.2. Dataset

The dataset used in our experiment is taken from a real life experiment conducted in the houses of elderly residents over the age of 60 [16]. The activity dataset was obtained by Matsui et al. through an extensive research conducted over a period of two months, where motion and environmental sensors were installed in each of the houses. Along with that, a physical button was installed in each of the houses, and the residents were requested to press the button whenever they started and ended an activity [16]. The original dataset consists of activity recognition data from single as well as two-person households. For the purpose of this research, we selected only single resident households that were three in total. We use cleaned and collected data from the above-mentioned study, and consider that the activity recognition system is 100% accurate (we used ground truth labels of activities in the dataset as the output of the activity recognition method).

The daily activities of the elderly that we want to track and monitor are mentioned in Table 1. The original dataset, however, does not contain data related to the Medication activity. Similarly, we also wanted to include multiple activities related to frequent use of TV. To fulfill our desired dataset, we added aforementioned activities into the original dataset. The total period of experiment of the two-month study was longer than our intended experiment period of 10 days. Hence, we only selected data for a 10 day period from the available two months of data. We included data from the same time period section for all the three single-resident households.

We included some risky situations into the dataset based on the definition shown in Table 2. For the purpose of our research, we included only *low* and *medium* level risks. As defined, *none* risk indicates that there is no problem with the elderly. Hence, we do not need to alter the dataset for such risk, since they concur with the regular routine of the elderly. If the level of risk is *high*, it indicates that the elderly person is in a serious condition and in need of immediate medical care. In such cases, no activity will be completed by the elderly, and the activity report in the application will not be updated.

However, our aim is to determine if any deviation from regular routine of the activities could be determined using our application. Though *high* risks can occur suddenly, we also think that, if we regularly monitor and determine *low* and *medium* level risks, then *high* level risks can be prevented or predicted. Because of this, we did not include *high* level risks in our dataset.

### 4.3. Experiment Details

We recruited a total of nine participants (gender: 6 Male, 3 Female; age range: 25–34 years old, average age: 28.6 years) to take part in our evaluation study. The participants were playing the role of ‘Monitors’ throughout the experiment. The modified dataset of the three single-person households were used for the three ‘Targets’ in the application. The participants were divided into three groups each. Thus, we had three participants each in three study groups. This was carried out to implement random distribution of our application in a way that each group, with an equal number of participants, will use a different application at a given time compared to other study groups. To implement that, we divided the experiment period into three phases in total. Table 4 simplifies the study group and application interface division.

The three versions of the application were uploaded to Google Play Store. Before the start of the experiment, we conducted a research and experiment introduction session that all the participants were requested to attend compulsorily. We explained the theme of the study and experiment in detail, their role as monitors, and the tasks they have to complete while using the application. They were also provided a document containing all the information about the working principles of the different versions of the application, along with QR codes for each version. The documents also indicated the version of the application they were supposed to use in each phase of the experiment. As a reward for participation in the experiment, the participants were provided with a gift card worth 2000 JPY.

To make the transition between interfaces easier for the participants, we included a one day gap between each phase. The participants were asked to take a break for a day in between the phases. The phases were designed to be of three days each. However, at the start of phase 2, we encountered some complications with the server connected to our application, and the application did not work properly until mid-day. Hence, we asked the participants to continue phase 2 for one day more. Thus, in total, the experiment period consisted of 12 days, with breaks of two days in total. After the end of each phase, we asked the participants to fill in a questionnaire developed using Google Forms. Most of the questions had to be rated on a five-point Likert scale (1 = strongly disagree, 3 = neutral, 5 = strongly agree), while some of them were open-ended. The participants were asked to respond to questions or statements related to their perception of the version of the application, as well as the effect of change in the version of the application, such as “*The activity related notifications were helpful in monitoring the elderly as it reminded me to check the application regularly.*”, “*I found the change in the interface confusing.*”, and “*I feel the new interface needed more mental effort.*” At the end of the experiment, the participants were asked to fill out a final questionnaire. The purpose of these questionnaires is to gain insight into the impression of the participants for different versions and different notification types.

### 4.4. Results

The results of our study are analyzed based on the following three conditions:Accurate detection of risky situations;Low burden of monitoring on Monitors;Timely Detection of risky situations.

#### 4.4.1. Accuracy of Risk Detection

In order to verify the effectiveness of our visualization technique, it is necessary to check if the risks included in the application, as mentioned in Section 4.2, will be identified correctly. In this section, we report the rate with which the risks included in the dataset were correctly identified in each phase, using different versions of the application. Table 5 and Table 6 show rate of correct identification of risks based on study groups and interfaces, respectively.

From Table 5, we can observe that StudyGroup C was the most consistent group, with the highest risk identification rate during all of the three phases of the experiment. The rate of correct identification also increased along with the experiment, which proves that familiarity with the application helped to analyze the activity reports and submit reports.

There was a slight decrease in risk identification for StudyGroup A when the interface changed from graphical (GAR) to tabular (TAR) in phase 2 of the experiment. All of the participants in StudyGroup A agreed that the new interface needed more time to analyze in their questionnaires after phase 2, with 66.7% agreeing that the tabular interface (TI) needed more mental effort than graphical interface (GI). When the interface changed to graphical layout (GR) in phase 3 of experiment, there was an increase in the correct rate identification. When asked about the change, participants claimed that it was easier to understand the routine with the graph compared to tabular layout (66.7% agree, 33.3% strongly agree).

StudyGroup B showed a considerable increase in correct risk identification, in phase 2, as shown in Table 5, even though they had graphical layout for both phase 1 (GR) and 2 (GAR). We can predict that familiarity with the application was the reason for such change. In their questionnaire after phase 2, 66.7% strongly agreed that they were familiar with the application and found it easier to use the application during this phase. However, in phase 3, their interface changed to tabular layout (TAR). This led to reduction in risk identification, with 33.3% strongly agreeing that the change in interface was confusing.

As shown in Table 6, we found out that, in total, using GAR, on average about 75.2% of the time the risks were identified correctly. In comparison, the risks were identified correctly about 65.8% of the time using TAR. GR, which in this context, is the same in visualization as GAR had a risk identification accuracy of about 68.5%. The average rate of risk identification is lower for tabular interface (TI), compared to both of the graphical interfaces (GI). This can help to identify that graphical interfaces (GI) provide better understanding or identification of risks.

We also found statistically-significant differences between the average risk identification rates of the three interfaces using the one-way ANOVA method (*p* = 0.037). A Tukey-HSD post-hoc test revealed a significant pairwise difference between interfaces GAR and TAR (*p* = 0.032) whilst no difference was observed between GAR and GR (*p* = 0.2).

To investigate this further, we combined the results of GAR and GR into a single group and compared it with TAR, to clearly determine differences between graphical and tabular interfaces for risk identification. Through the paired *t*-test analysis, we found that there is a significant difference between the two (*p* = 0.047).

#### 4.4.2. Low Burden Evaluation

We define burden as the time taken by the participants between opening the application to check the activity report of targets and submitting the report. We logged the time of opening of the application as well as the time of reporting using “Shared preference” functionality available for Android developers. These time periods were saved together in the Firebase database. We analyzed the burden time for each participant using this data and calculated an average burden time for each participant over the whole experiment period, which is shown in Figure 8. The average burden time for each of the versions is also shown.

We can see that the burden time for GAR, on average, is always less than TAR. The mean burden time for GAR, TAR, and GR were observed to be 28 s, 38 s, and 52 s, respectively. As seen in Figure 8, the burden for participant 1 while using GR is very high compared to other participants, and other interfaces used by the same participant. Upon inspection, it was discovered that, while using GR, for one particular report, the participant recorded an unusually high burden time, which was uncharacteristic for the participant based on his other responses. Discarding the unusually high burden time, the average burden time of the participant 1 was reduced from 193 s to almost 20 s. However, for the final analysis, the skewed data are kept as it is. Similarly, the burden for participant 2 while using TAR is zero because the participant did not record any response during phase 2 of the experiment.

To analyze the link between burden of using the application, and engagement with the application over time, we calculated the average time it took to report based on the phases of the experiment. The results are shown in Figure 9. When the interface changed from graphical (GAR) to tabular (TAR), in phase 2 for StudyGroup A, we can see that the burden time was higher. In phase 3, when their interface changed back to graphical (GR), the burden time was observed to be extremely high (94 s) due to the unusual reporting by participant 1 as explained above. Discarding that particular incident, the burden time was observed to be lower than in phase 2 (28 s).

For StudyGroup B, the burden time was highest in phase 1, with 47 s, when using GR. However, the burden time decreased in phase 2 (25 s) when using GAR. This can be attributed to the participants getting familiar with the interface. In phase 3, however, when the interface changed to tabular (TAR), we can see that the average burden time increased to 37 s.

Similarly, when the interface was changed from tabular (TAR) to graphical (GR), for StudyGroup C in phase 2 of experiment, we can see that the average burden time was lower (22 s). Even though the burden time increased in phase 3 (25 s), using GAR, it was still lower than the burden time in phase 1 (42 s). Therefore, over the course of the experiment period, we can observe that change in interface had some effect on the engagement with the application and burden time. Familiarity with the application lowered the burden time, especially using a graphical interface (GI).

We found a statistically-significant difference in the burden time for the three interfaces using a one-way ANOVA method (*p* = 0.012). A Tukey-HSD post-hoc test revealed a significant pairwise difference between interfaces GAR and TAR (*p* = 0.039) whilst no difference was observed between GAR and GR (*p* = 0.13).

For further investigation, we combined the results of GAR and GR into a single group and compared it with TAR and through a paired *t*-test analysis; we found that there is a significant difference between the two (*p* = 0.049). This analysis, along with the results from Figure 8 and Figure 9, help to show that there is a significant difference between tabular and graphical interfaces for the burden faced while using the application, with a graphical interface resulting in a lower burden for the participants.

Lesser burden also resulted in higher engagement with the application. Figure 10 shows that the total number of reports received using GAR across different phases were almost consistent across the three phases, and on average higher than when using TAR. There was a significant decrease in reports using TAR in phase 2 for StudyGroup A. This can be attributed to change in their interface because, in an earlier phase, they used graphical interface (GI). They also mentioned in the questionnaire after phase 2 that tabular interface (TI) was difficult to understand, which resulted in a lower number of reports.

We can thus conclude that GAR provides lesser burden to participants, in comparison with TAR, and on average has higher engagement and reporting. This further strengthens our proposal that graphical interface (GI), with adequate textual information, can be helpful for monitors to identify the routine of targets and distinguish risky situations whilst spending less time and effort analyzing the interface.

#### 4.4.3. Timely Detection

Figure 11 shows the time taken to report about a completed activity during each phase, based on types of interface. Over the three phases of experiment, we can observe that using a graphical interface (GI), the reports for activities were received quicker compared to tabular interface (TI): GAR (average = 176.46 min, median = 115.01 min), TAR (average = 201.42 min, median = 118.85 min), and GR (average = 166.9 min, median = 121.12 min). Even though such high response times for the report are not favorable, we think that there were many factors that affected the reporting time for activities.

The time of notification generated, which is also the time when the activities were completed, was saved using "Shared preference" functionality, as mentioned in Section 4.4.2. Similarly, we also saved the time when the activity report was submitted. We determine the time taken to report an activity by calculating the time difference between report submission and notification generation. For StudyGroup A, when the interface changed from graphical (GAR) to tabular (TAR) in phase 2, the reporting time was higher compared to phase 1, even if they had received both rN (recurring notifications) and abN (activity based notifications) in both of the phases. This can be attributed to the change in interface because, when their interface changed back to graphical (GR) in phase 3, the time of response also was observed to be lower than on phase 2, even though they did not receive abN. This shows that type of visualization can have an effect on the response time for notifications received.

StudyGroup B were almost consistent in their performance throughout the first two phases of the experiment period. In phase 2, when their interface changed from GR to GAR, there was no significant change in their response time even if they did not receive abN. However, when their visualization changed to tabular (TAR) in phase 3, the time of responses was higher than in the previous two phases.

In contrast, StudyGroup C did not show any significant differences in response time for activities based on changes in interface as well as reception of abN. When their interface changed from TAR to GR in phase 2 and from GR to GAR in phase 3, their response time for notifications did not show any high amount of significant differences. StudyGroup C thus did not show any conclusive effect for the change in visualization or notification strategies for the reception of reports to activities.

Table 7 shows the average response time of each participant while using each of the interfaces, where the lowest response time taken among the three interfaces is highlighted. Even though TAR consisted of both abN and rN notifications, we found that none of the participants responded quickly while using it. Moreover, the mean response time using TAR is highest across all the participants (except participant 2, who did not register any response during phase 2). We found that, even though they did not receive abN, some of the participants (4) recorded lowest mean response time using GR. GAR and GR recorded mean response times of about 176.46 min and 166.9 min respectively, while TAR had a mean response time of 201.42 min. Even though GR had lower average response time, we observed that the median response time for notification was lower for GAR (115.01 min) compared to GR (121.12 min) and TAR (118.85 min). This shows that reports were received quicker using GAR than GR or TAR.

Upon further analysis, we found statistically-significant differences between activity response time for the three interfaces using a one-way ANOVA method (*p* = 0.005). A Tukey-HSD post-hoc test revealed a significant pairwise difference between interfaces GR and TAR (*p* = 0.05) whilst no difference was observed between GAR and GR (*p* = 0.64) or between GAR and TAR (*p* = 0.055).

We then combined the results of interfaces that received abN, i.e., GAR and TAR, into a single group and compared it with GR, and found that a paired *t*-test shows a significant difference between the two (*p* = 0.022).

This shows that reception of abN does indeed have an effect on the time for response to the activities. To investigate this further, we determined the time range within which the responses to the activity notifications were received. Table 8 shows the cumulative percentage of reports received within the given time ranges for the three versions of the application. We divide the time into 30 min intervals; however, the table only shows until 210 min, since the highest average time of response is within the 180–210 min range. We can see that the amount of responses received does not vary by a large amount if graphical interfaces are compared. However, for tabular interfaces, the response rate is lower even if abN was received. This shows that abN, when used with a graphical interface, provides a better result than compared with tabular interface. We then tried to investigate which interface provided the quickest response for activities.

We divided the notifications into those that were for regular activities and those that were for the risky situations. By using the time taken to report to activities, we determined the minimum time taken to submit a report for an activity among all the participants, and the version of the application used to submit that report. Thus, we found that, using which particular version of the application, we received the quickest response for each of the activities. The results are shown in Figure 12 and Figure 13. We can see that the risky situations responded quicker when using interfaces that consisted of abN, even though there is not much difference between interfaces for the quickest time of response to non-risky notifications.

In the final questionnaire, the participants responded with the reasons that could also provide the reason for such higher response time. Almost 45% participants (*n* = 4) mentioned that they were busy with their research/private work and could not respond to the notifications on time. We received responses such as: *“I was so busy with my work”; “Busy with my research work or play a game"; “mentally busy with my own work"; “sometimes i was busy”*. Similarly, two of the participants mentioned that they often forgot to check the application. This can be attributed to the different interface types used and notifications received.

Two of the participants responded in the questionnaire that they did not use the application if they did not receive any notifications, while six (66%) of them said they did not wait for the notifications to use the application but were busy with their work and could not respond immediately. We also wanted to know if the notifications received were perceived as distracting or disturbing, to analyze if their perception played any role in the response time. When asked if the notifications received from the application were distracting, 2 (22%) of them strongly claimed they were not disturbed, 5 (55%) said they were not disturbed, while 1 of them was neutral, and 1 agreed that he was distracted. Similarly, 8 (88%) (strongly agree: 4; agree: 4) agreed that they prefer to receive abN so that they can be regularly notified monitor frequently, while 1 of them was neutral.

## 5. Discussion and Limitations

In this section, first we discuss the results and verify research questions RQ1–3 mentioned in Section 2.3, then we show some remaining issues as limitations.

### 5.1. Discussion

When considering user engagement and their ability to identify routine of individuals with the interface, we can conclude that the results are fairly positive towards GAR, as compared to TAR. Using GAR, we found that 75.2% of risky situations were correctly identified as risks, compared to 65.8% and 68.5% for TAR and GR, respectively. Though identification of risk varied between study groups using GAR (68.4% for StudyGroup A; 64.7% for StudyGroup B, and 92.6% for StudyGroup C), the overall identification rate is higher for GAR. This shows that risks can be identified using graphical interface and the style of graph that we used. A response from a participant, *“I can see the difference of the duration directly from the graph. The table one need to scroll up and down to see all the information, which sometimes kind of annoying”* also suggests that our visualization is effective. These findings justify our research questions, RQ1 and RQ2, that it is possible to identify the daily routine of individuals using a smartphone application, and it is possible to detect potential risks in such routine based on the visualization provided.

Using GAR, participants faced the lowest burden of 28 s, compared to 38 s in tabular (TAR). Similarly, none of the participants claimed that the application demanded a lot of time and effort from them. Regarding notifications, only one of the participants found them distracting, and 88.8% mentioned that they will prefer to receive activity based notifications for monitoring purposes. Similarly, all of the participants (77.8% strongly agree, 22.2% agree) responded that the use of traffic colors was useful to identify the state of the activities quickly. Therefore, we can verify RQ3, that constant notifications and using the application was not troublesome for the users.

We received a total of 1680 responses from participants over the experiment period. We can claim that such interaction is a result of their willingness to use the application. When interface of participants changed from graph to table, there was a reduction in the number of reports obtained (45.6% for StudyGroup A in phase 2, and 9.8% for StudyGroup B in phase 3). Similarly, when the interface changed from tabular to graph, we obtained an increase in the number of reports by 96.7% for StudyGroup A in phase 3 and reduction by 11.5% for StudyGroup C in phase 2. In total, the engagement with the application is high, which along with the lower interface analyzing time, verifies RQ3, that using the application is not a burden for the monitoring person.

At the end of the experiment, we asked the participants which representation of activities they preferred: table or graph. All of them agreed that graphical representation was better. Some of the responses we received, such as, *“Got on a quick glance the exact duration of past activities and could check exact time of the day”; “With graph, it’s easy for me to compare the length of activity at the glance.”*, further strengthens our proposal that the graphical interface we proposed can help to identify a daily routine in a clear and intuitive manner and further justifies RQ1, that a smartphone application can be a good tool for identifying daily activities.

### 5.2. Limitations

Our system evaluation requires that there are certain risky situations in the activity of the elderly. We did not conduct a real-time activity recognition of elderly, but instead, we used a pre-existing activity dataset because, in real-time scenarios, there is no surety of receiving such risky situations, and we would need to request someone to deliberately change their activity pattern so that others could detect it. Such a situation can invoke unfavorable reactions. Similarly, since activity recognition systems are not perfectly accurate, sometimes the activities may not be correctly identified, or falsely identified, which would hamper our evaluation. Moreover, we recruited students for the experiment, but they are always busy because of their academic work, and/or personal lives which might have affected the number and time of reception of reports.

## 6. Conclusions

In this study, we proposed a system, PATROL, that can be used to anonymously track everyday activities of the elderly and identify any potential risks in their daily routine using a smartphone application. Our system is aimed to be deployed in elderly residential areas or communities and does not disclose any private information such as age, location, etc. to the monitoring person to maintain the privacy and security of elderly residents. The monitoring person receives recurring notifications every two hours and activity-based notifications whenever an elderly person completes an activity from the service server and assesses elderly condition by a smartphone application visualizing elderly activity history. We designed our application with features such as single interface design, intuitive graphical user interface for activity and anomaly detection, and color and textual information for state of activities. These features altogether help not only to conduct quicker monitoring of activities of elderly, but also to induce a low amount of burden to the monitoring person, who at once may be responsible for monitoring single or multiple elderly people.

We added risky situations in an activity dataset obtained from a real-life experiment with elderly residents and conducted a user study using the proposed method and two other baseline methods varying in visualization and notification techniques for three groups consisting of nine participants. We found that with our proposed method, 75.2% of the risks were successfully identified, while 68.5% and 65.8% were identified with other methods. The proposed method also provided a better result for the timely reception of activities: GAR ( median = 115.1 min), TAR (median = 118.85 min), and GR (median = 121.12 min). Moreover, the interface analyzing and reporting time was also lower (28 s) in the proposed method compared to 38 and 54 s in other methods. As future work, we will conduct real-time activity recognition and monitoring using our application. To achieve that, we will also research/work on activity recognition systems using other kinds of sensors that can not only potentially provide better activity recognition in real time but also remove dependency on the elderly person for data collection. Moreover, we will explore the possibility to assess the elderly’s activity state and detect anomalies by using measurements from ambient sensors (temperature, humidity, illumination, etc.). We will also include high risk situations such as Fall (and no activities after the incident) and try to determine if participants will be able to deduce such emergency situations quickly. We will also aim to increase the number of participants to receive more reports and analyze the results based on age, gender, etc.

## Figures and Tables

**Figure 1 sensors-22-06965-f001:**
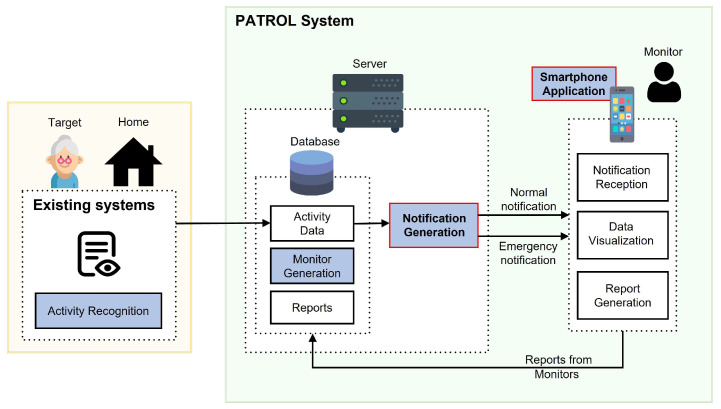
System architecture of PATROL.

**Figure 2 sensors-22-06965-f002:**
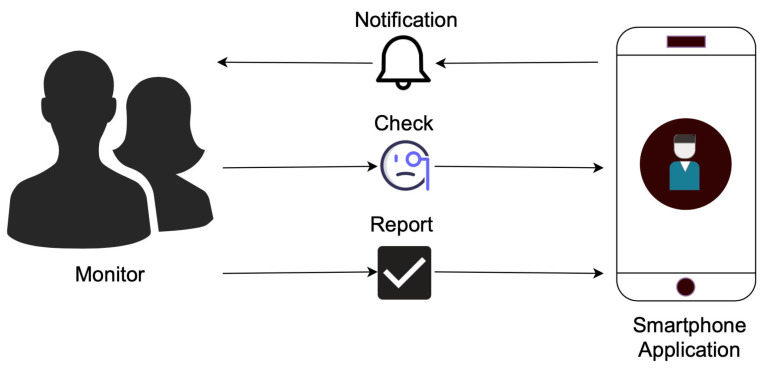
Interaction between Monitor and smartphone application.

**Figure 3 sensors-22-06965-f003:**
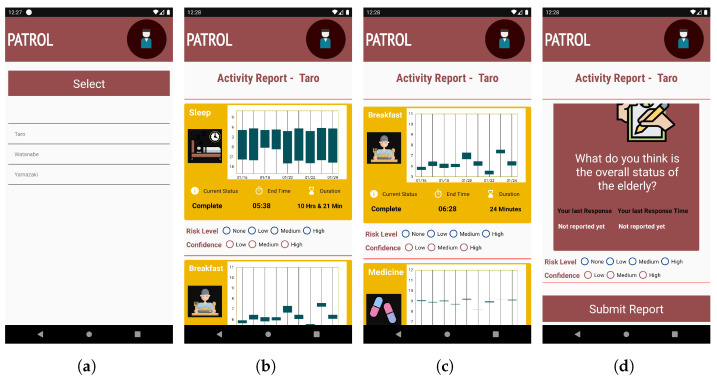
Snippet of the smartphone application for: (**a**) choosing Targets, (**b**) sleep card, (**c**) breakfast card, and (**d**) submitting report.

**Figure 4 sensors-22-06965-f004:**
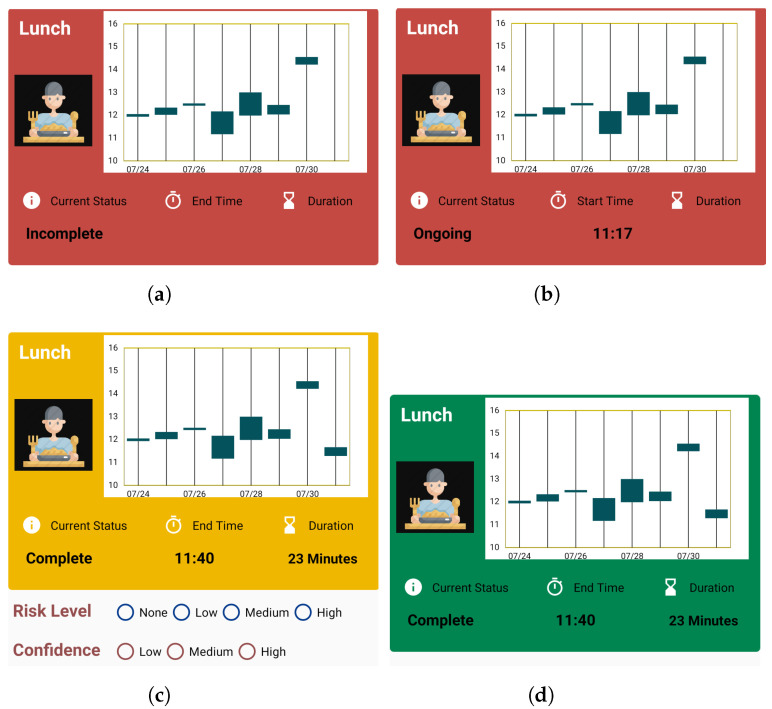
Use of color for representing activity state for: (**a**) activity not complete, (**b**) activity ongoing, (**c**) activity complete, and (**d**) activity reported.

**Figure 5 sensors-22-06965-f005:**
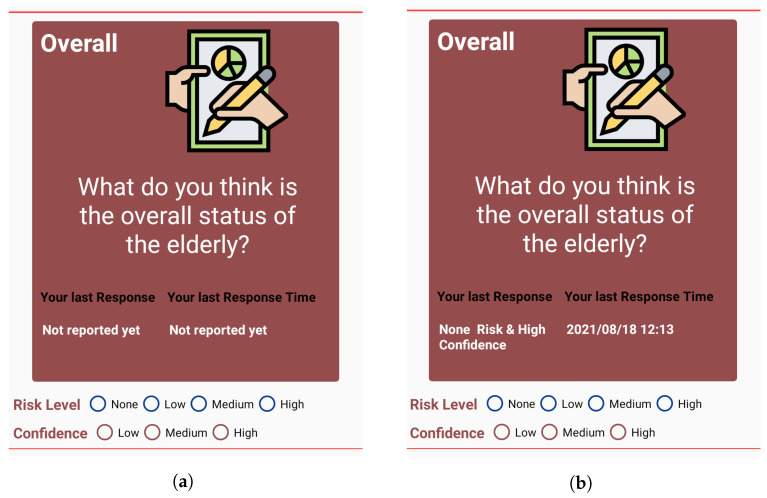
Overall report card: (**a**) before report submission and (**b**) after report submission.

**Figure 6 sensors-22-06965-f006:**
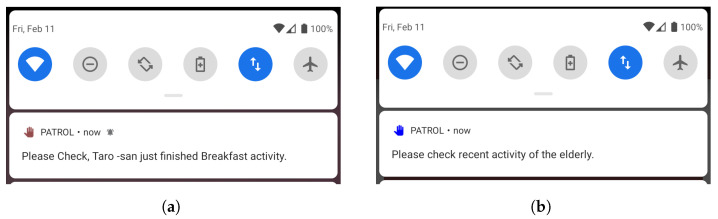
Example of notifications generated: (**a**) activity based notification (abN) and (**b**) recurring notification (rN).

**Figure 7 sensors-22-06965-f007:**
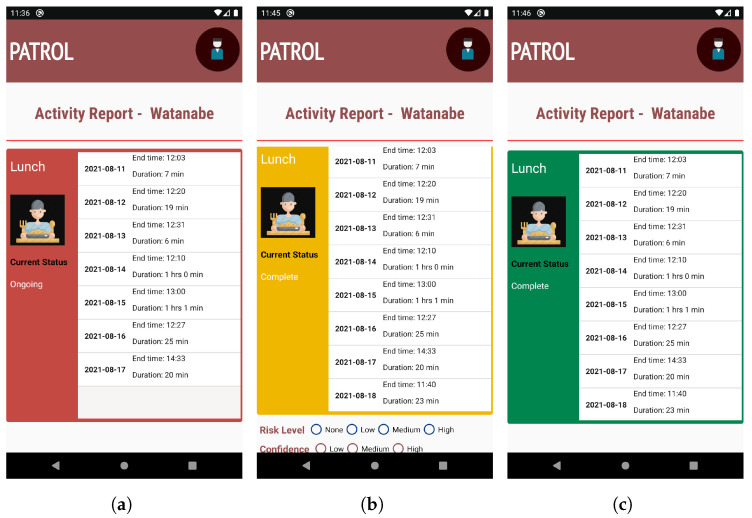
Example of tabular interface (TI): (**a**) activity incomplete, (**b**) activity complete and (**c**) activity reported.

**Figure 8 sensors-22-06965-f008:**
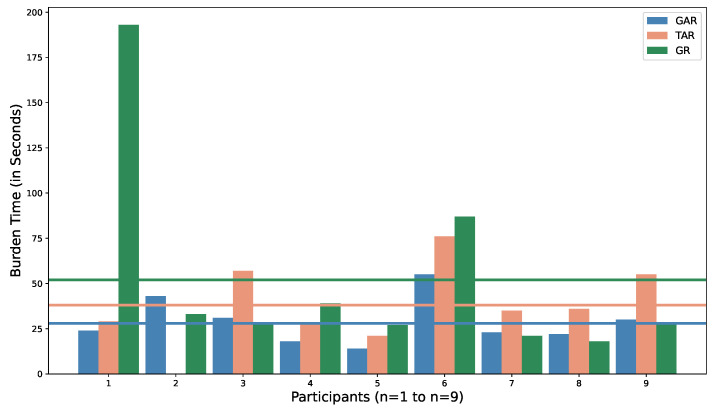
Average burden time of participants.

**Figure 9 sensors-22-06965-f009:**
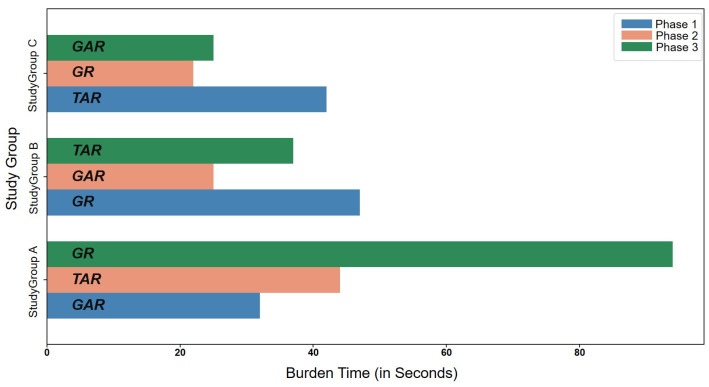
Average burden time of study groups per phase.

**Figure 10 sensors-22-06965-f010:**
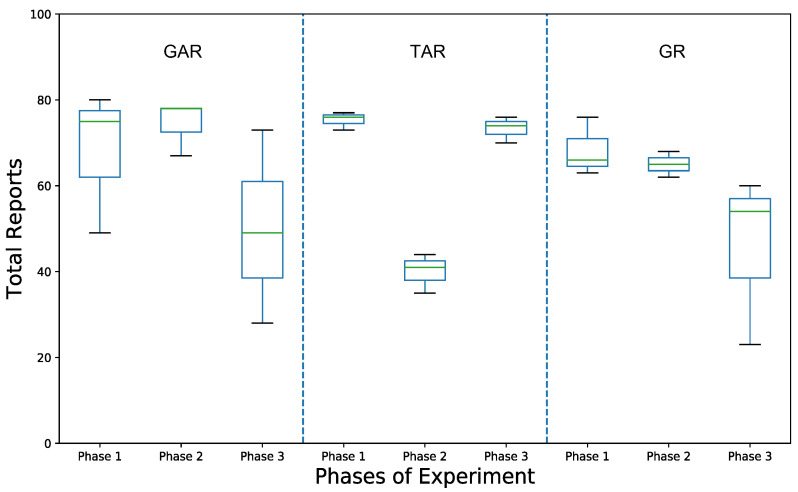
Total number of reports received.

**Figure 11 sensors-22-06965-f011:**
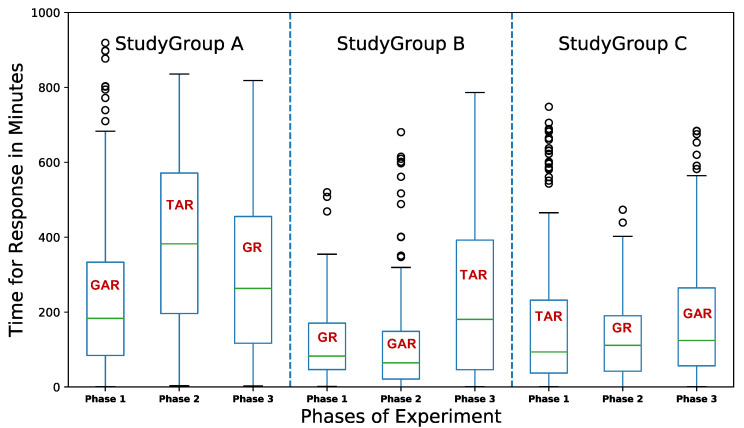
Response time for activities per phase based on study groups.

**Figure 12 sensors-22-06965-f012:**
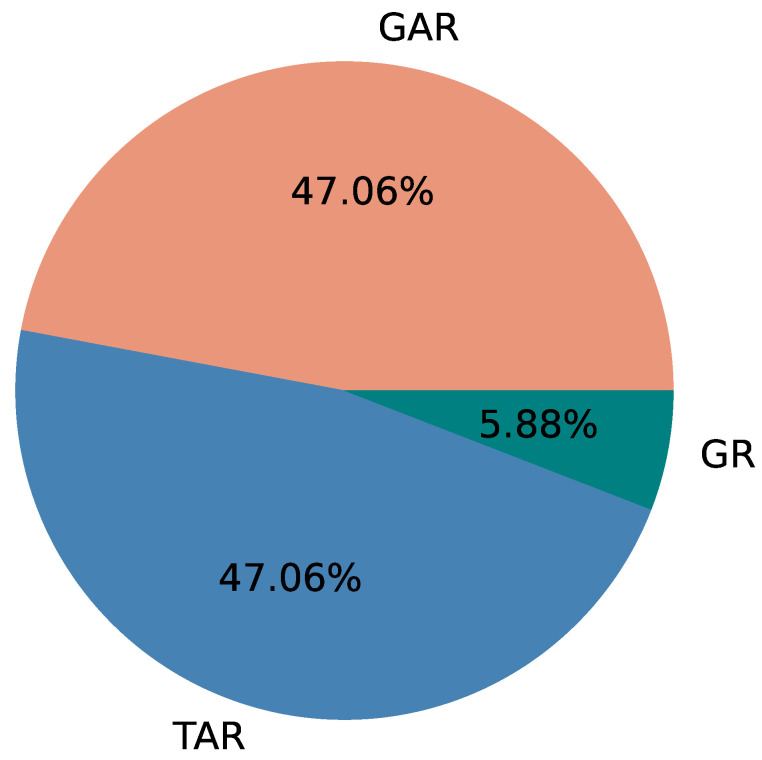
Quickest response for risky situations.

**Figure 13 sensors-22-06965-f013:**
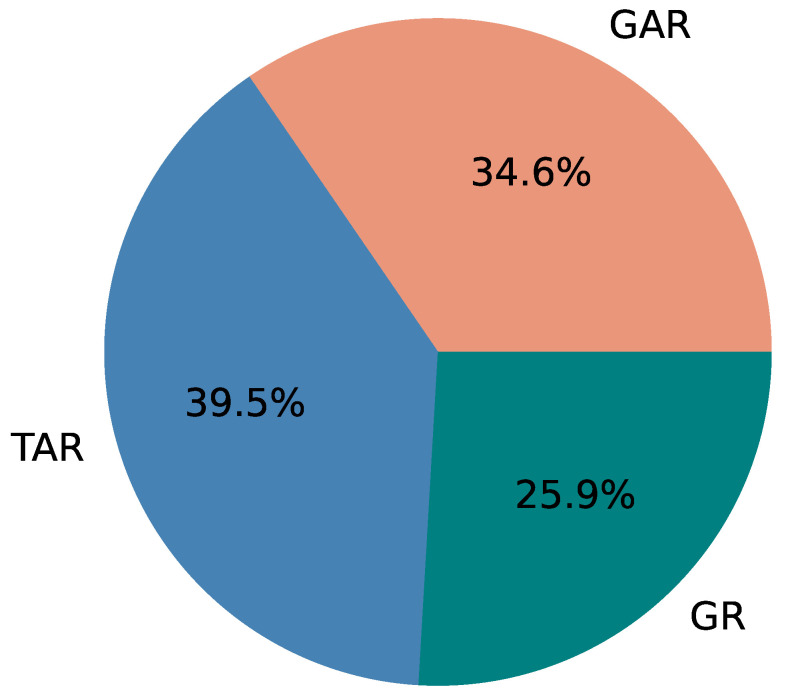
Quickest response for non-risky situations.

**Table 1 sensors-22-06965-t001:** Areas and activities to monitor.

Area	Home Objects with Sensors	Activities
Kitchen	Stove Microwave	Breakfast, Lunch or Dinner
Bedroom	Bed Medicine Bottle	Sleep time Medication
Living room	TV	Entertainment
Bathroom	Water consumption	Shower

**Table 2 sensors-22-06965-t002:** Risks used in the application and their description.

Risk	Description
None	Everything seems to be okay with the elderly.
Low	There is some problem, but can be handled by elderly themselves.
Medium	There is a problem and the elderly should be assisted/checked by caretaker, family members or doctor.
High	Elderly is in emergency and requires immediate medical care.

**Table 3 sensors-22-06965-t003:** Types of versions of PATROL application.

Version	Interface		Notification	
	Graphical	Tabular	Activity-Based	Recurring
GAR	✓	–	✓	✓
TAR	–	✓	✓	✓
GR	✓	–	–	✓

**Table 4 sensors-22-06965-t004:** Study groups and division of version of PATROL application.

Phase	Date (MM/dd)	Number of Days	StudyGroup A	StudyGroup B	StudyGroup C
1	08/25–08/27	Three	GAR	GR	TAR
2	08/29–09/01	Four	TAR	GAR	GR
3	09/03–09/05	Three	GR	TAR	GAR

**Table 5 sensors-22-06965-t005:** Risk identification based on study groups.

Study Group	Phase	Interface	Risk Identification	Average
StudyGroup A	1	GAR	68.4%	72.6%
	2	TAR	67.7%	
	3	GR	81.9%	
StudyGroup B	1	GR	32.4%	46.1%
	2	GAR	64.7%	
	3	TAR	41.3%	
StudyGroup C	1	TAR	88.4%	90.7%
	2	GR	91.3%	
	3	GAR	92.6%	

**Table 6 sensors-22-06965-t006:** Risk identification based on interface types.

Phase	GAR	TAR	GR
1	68.4%	88.4%	32.4%
2	64.7%	67.7%	91.3%
3	92.6%	41.3%	81.9%
Average	75.2%	65.8%	68.5%

**Table 7 sensors-22-06965-t007:** Mean response time (in minutes) of each participant.

Participant	GAR	TAR	GR	Total Average
1	**225**	349	243	272
2	**335**	No response	341	225
3	**182**	418	260	286
4	146	287	**135**	189
5	**33**	146	59	79
6	**154**	301	167	207
7	187	188	**146**	173
8	114	171	**109**	131
9	205	146	**110**	156

The quickest mean response time for each participant is highlighted in bold text.

**Table 8 sensors-22-06965-t008:** Cumulative percentages of responses received per time range (in minutes).

Time Range	GAR	TAR	GR
0–30	18.4%	15.45%	13.24%
30–60	30.55%	23.21%	29.66%
60–90	42.36%	31.22%	41.79%
90–120	51.56%	37.09%	49.44%
120–150	57.63%	41.42%	58.39%
150–180	63.88%	44.04%	65.29%
180–210	68.92%	47.29%	70.52%

## Data Availability

The data presented in this study are available on request from the corresponding author.

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
