# Peer review of "PATROL: Participatory Activity Tracking and Risk Assessment for Anonymous Elderly Monitoring"

_sensors, 2022, doi:10.3390/s22186965_

Round 1

Reviewer 1 Report

Review Comments and Suggestions

Title: PATROL: Participator Activity Tracking and Risk Assessment for Anonymous Elderly Monitoring

Journal: Sensors

Manuscript ID: 1825120

Reviewer Comments

Authors have proposed a Monitoring System that tracks the activities of Elderlies remotely, using existing ‘sensors’ and a new ‘smartphone application – PATROL’. Though the remote tracking of household activities is entirely not new, the approach proposed in this study is a good initiative to reduce cost while still maintaining the effectiveness in tracking routine activities. In this study, routine activities of Elderlies are remotely tracked by ‘Monitors/Caregivers’ using PATROL. Three methods of visualization for activity tracking were tested in this study, of which GAR (Graphical Interface) was found to be less burden on the ‘Monitor’ compared to TAR (Tabular Interface). Considering the study involved 10 days of Elderly monitoring with 9 participants across 3 methods of PATROL visualizations, this is kind of a pilot study, and the authors should include real-life scenarios in future to make the PATROL competitive and beneficial. This study should address the below comments for further consideration of publication.

Page 1

Lines 19 – 20: Authors needs to update the statistics as there are more recent surveys.

Page 2

Lines 52 – 53: Article Title doesn’t sync with the Acronym stated here.

Line 65: Check the Grammar.

Lines 64 – 65: It’s not made clear whether the smartphone application and notifications described in here is with respect to elderly people, remote caretakers, or relatives, while the statement it follows covers both perspectives of self- and monitoring person. In addition, how to address the privacy issues once the caretaker of a monitoring person has left the role?

Lines 72 – 73: How do you define ‘Anonymous Monitoring’, when elderly is knowledgeable of this monitoring? And how shall it protect the privacy of elderly?

Lines 74 – 77: From Author’s perspective, any deviation from start/end time and duration of any activity shall be considered a RISK. But in this study, it wasn’t mentioned how it will be dealt if there are cases of (1) large deviations, (2) intermittent activity periods, and (3) non-entries (like physical button non-press). And how the same will be addressed in future experiments/ mass testing?

Page 3

Lines 83 – 84: Please check the Grammar.

Lines 87 – 92: Text seems redundant from the previous page. Authors can improve the context of this summary.

Lines 93 – 97: Please check the Grammar.

Page 5

Lines 216 – 219: It is not made clear how ad-hoc tasks or non-routine tasks are handled?

Page 7

Table 1: One of the stated scopes is to detect “Fall”. But there is no mention about this.

Lines 264 – 265: Are these notifications same as that mentioned in Page 3? If so, General notifications will be sent once every 2 hours, and Emergency notifications soon after the task? But the narrative in section 3.1.3 is different. Or The General notification is subdivided into ‘2-hour notifications’ and ‘Activity-based notifications’? Please clarify.

Line 267: ‘Emergency notifications’ are the ones that is necessary in Elderly care. But focusing more on Monitor’s response-ability, the actual purpose of this study seems suppressed.

Lines 277 – 279: As initial assessments of activity tracking are mandatory to identify a pattern, who does this assessment – Monitor or System Administrator? How this will be handled where there are changes in Monitors or System Administrators?

Page 8

Lines 287 – 289: What’s the approximate time taken by the Monitors from the point of Actual risk time to the report submission, considering the Elderly people living alone needs quicker attention in case of emergency situations compared to routine tasks. Are Monitors allowed to contact medical response teams in case the emergency contacts don’t respond – How shall such situations be handled?

Lines 295 – 302: This is another example showing more focus on “lesser burden to Monitor/Caretaker” rather than “Priority to Elderly Care”. There should be a balance between these two to make the PATROL beneficial.

Page 9

Lines 322 - 325: For tasks that can occur multiple times in a day, should the Monitor need to manually input these records and submit as one report or how? Please clarify.

Lines 335 – 338: Retaining red color background for ‘Ongoing’ activities seems not a good choice. Considering the task has already started, it would have been more suited had the color of the activity card is yellow, but without the radio buttons for status reporting.

Page 10

Lines 365 – 368: So far it hasn’t been clear how a risk is considered low, medium, high? I understand Table 3 mentions about this – but this is completely not clear, especially when differentiating routine versus emergency situations.

Page 11

Lines 370 – 371: For Confidence levels, what are the parametric used to define low, medium, high when there are multiple Monitors monitoring Elderly at different times/days.

Line 382: “To provide context of the notification to the targets” – Shouldn’t it be to the Monitor?

Page 12
Line 401: Shouldn’t it be directed to Monitors instead of Targets?

Line 402: What is GAR, TAR and GR? Do they correspond to Versions 1, 2 and 3 of the PATROL application? They should be defined instead of letting the audience know their meanings from Table 2.

Page 13

Lines 406 – 408: Authors are encouraged to provide information about accuracy and reliability of these motion and environmental sensors used in 2-month survey. On other hand, the actual number of days of monitoring elderly seems relatively low (as stated in Table 4) – hardly 10 days – 3 to 4 days each for version testing.

Lines 408 – 409: Were the residents provided with any questionnaire or feedback form or any other quality control measures that were considered about confirming the accuracy of the stated action. It will have direct impact on the methods used for dataset generation as stated in lines 432 – 440.

Line 411: Only 3 households (with single resident) were considered for dataset generation? Was the risk level assessment (stated in lines 435 – 440) were based only on these 3 households over a period of 3 months? What differences were noticed from that of the two-person households if the latter is still being operated over the 2-month period?

Line 423: Another instance where ‘emergency task’ is omitted. Though the lines 429 – 431 does speak about the reason for non-inclusion of ‘high risk level’ in the dataset, Authors should have included it to determine the frequency of ‘high risk’ episodes experienced by the elders and subsequent actions to be/taken by Monitor/System Administrators.

Page 14

Line 444: It should be made clear that the 9 participants here are playing the role of ‘Monitors’.

Lines 452 – 457: Are there any stated prerequisites (knowledgeable) for the 9 participants as they must understand and experiment with the application?

Lines 463 – 468: What sort of questions were provided in the questionnaire to the 9 participants? Other than Application-related, were there any questions about elderly response actions (physical buttons), motion/environmental sensors – as these items are related feedback over application-related tasks.

Page 15

Lines 484 – 498: Totally agree with Author’s observations that Graphical Interface is most preferred over the Tabular Interface. But it should be noted that the 3 participants in each group were allowed only once to the Tabular interface compared to Graphical Interfaces. Moreover, the TAR method registered lower ‘Risk Identification’ when the StudyGroups have moved from Graphic to Tabular Interface resulting in a large deviation, but this is insignificant when the StudyGroup has moved from Tabular to Graphic interface(s). In addition, Tabular Interface provides accurate periods of activity/risk while the Graphical Interface provides a range. These scenarios will have an impact on Monitor’s Burden Evaluation as shown in Section 4.4.2.

Page 18

Section 4.4.3: Does the Time Taken to report an Activity is from the (1) time of notification, or (2) time of opening the Application? Though the lines 578 – 580 does mention about this, it is still unclear.

Page 21

Lines 661 - 663: It should be noted that RQ1 and RQ2 is dependent on both ‘Elderly’ and ‘Monitor’, as the former should press the Physical Button, and then only the Monitor can track the activity. In future studies, more real-life scenarios should be included while dependency on Elderlies should be removed to make the PATROL more effective.

Author Response

Dear reviewer,

Thank you for your valuable suggestions to our paper. We have made changes to the manuscript based on your comments. Please check the attached file to view our response. 

The changes we have made in the manuscript are highlighted in blue color, and are accompanied by a todo note to refer to the suggestion.

Sincerely,

Research Dawadi

Reviewer 2 Report

The authors present a monitoring method and its efficiency observing different activities through a distant application in order to detect risks in day activities. The subject is interesting and related work well presented, however the study has some major flaws.

- the number of people involved in the study seams not sufficient

- the fact that the information is analyzed after pressing a button and not directly obtained from sensors limits the number of false information, so the number of notifications. Thus, the rest of the study and conclusions may be strongly different.

- the choice of the characteristics of the 3 versions of the application is not clear and seams hazardous. Actually, the results of comparing a tabular version and graphic one is quite predictable. Then comparing recurring notification and activity ending one is interesting but in this case,  it would have been preferable to analyze a version where only activity ending notification is done. In addition, time between two notification for the recurring notification should be an interesting parameter which is not explored.

Author Response

(The authors gave the same response as above.)

Round 2

Reviewer 1 Report

The authors have addressed the comments satisfactorily, except for One. In addition, I have one more comment. Two minor comments that I have are:   1. Line 89: "part-time civil servants committed by Minister of Health, Labour and Welfare" needs revision in clarifying whether it refers to "commissioned by Ministry on a voluntary basis" or "mandatory participation of part-time civil servants as volunteers"   2. Line 111: Grammar check remains unresolved for the phrase "....notifications in the smartphone application to trigger frequent use the application..."

Author Response

Dear reviewer,

Thank you for your valuable suggestions to the revised version of our paper. We have made changes to the manuscript based on your comments. Please check the attached file to view our response.

The changes we have made in the manuscript are highlighted in blue color, and are accompanied by a todo note to refer to the suggestion.

Sincerely,

Research Dawadi
